# Conserved ancestral tropical niche but different continental histories explain the latitudinal diversity gradient in brush-footed butterflies

Nicolas Chazot[1,2,3✉], Fabien L. Condamine [4], Gytis Dudas[3,5], Carlos Peña[6], Ullasa Kodandaramaiah [7], Pável Matos-Maraví [3,8], Kwaku Aduse-Poku[9], Marianne Elias[10], Andrew D. Warren[11], David J. Lohman [12,13], Carla M. Penz[14], Phil DeVries[14], Zdenek F. Fric[8], Soren Nylin[15], Chris Müller[16], Akito Y. Kawahara[11], Karina L. Silva-Brandão [17], Gerardo Lamas[6], Irena Kleckova[8], Anna Zubek [18], Elena Ortiz-Acevedo[11,19], Roger Vila [20], Richard I. Vane-Wright[21,22], Sean P. Mullen[23], Chris D. Jiggins [24,25], Christopher W. Wheat[15], Andre V. L. Freitas [26] & Niklas Wahlberg [2]

The global increase in species richness toward the tropics across continents and taxonomic groups, referred to as the latitudinal diversity gradient, stimulated the formulation of many hypotheses to explain the underlying mechanisms of this pattern. We evaluate several of these hypotheses to explain spatial diversity patterns in a butterfly family, the Nymphalidae, by assessing the contributions of speciation, extinction, and dispersal, and also the extent to which these processes differ among regions at the same latitude. We generate a time-calibrated phylogeny containing 2,866 nymphalid species (~45% of extant diversity). Neither speciation nor extinction rate variations consistently explain the latitudinal diversity gradient among regions because temporal diversification dynamics differ greatly across longitude. The Neotropical diversity results from low extinction rates, not high speciation rates, and biotic interchanges with other regions are rare. Southeast Asia is also characterized by a low speciation rate but, unlike the Neotropics, is the main source of dispersal events through time. Our results suggest that global climate change throughout the Cenozoic, combined with tropical niche conservatism, played a major role in generating the modern latitudinal diversity gradient of nymphalid butterflies.

A full list of author affiliations appears at the end of the paper.

Understanding the uneven distribution of biodiversity on Earth is one of the most fundamental goals in ecology and evolution. Numerous patterns of biodiversity distributions have been documented, but the obvious increase in species richness from the poles toward the equator known as the latitudinal diversity gradient (LDG) is remarkable for its consistency across geographic scales and taxonomic groups[1–3]. Although many different hypotheses have been formulated to explain this pattern, no consensus has emerged.

The proposed hypotheses fall into three broad categories: ecological, evolutionary, and historical[1]. The increasing availability of molecular phylogenies has renewed interest in evolutionary and historical hypotheses because they provide an opportunity to infer some of the past history without extensive fossil information[4–7]. Four historical processes that could result in greater species richness in tropical regions are usually proposed.

First, longer time-for-speciation in the tropics[8,9]. During the early Cenozoic, tropical biomes were found across much higher latitudes, while colder biomes with higher seasonality expanded only after the Eocene-Oligocene boundary[10]. Many groups from the early Cenozoic probably originated in these tropical areas. Assuming similar speciation and extinction rates across regions, species richness would therefore be greater in the tropics if lineages had more time to accumulate[9].

Second, asymmetric dispersal events between the tropics and other areas. Clades originated either in the tropics and rarely dispersed out of them[11] or instead originated in temperate regions and frequently dispersed into the tropics, thereby increasing tropical species richness (e.g., ref. [12]). The first scenario is expected in the cases where most tropical organisms that are highly specialized to their environmental niche cannot colonize different ecological conditions, such as those in seasonal temperate regions. Consequently, such colonization events may be rare and recent, resulting in strong conservatism of the tropical niche. The second scenario implies that clades originated in high-latitude (temperate) regions but colonized tropical regions frequently, resulting in repeated evolution of adaptations for tropical existence[13].

Third, higher speciation rates in the tropics ("cradle of diversity"[14]). Tropical lineages speciate more rapidly than temperate lineages (e.g., ref. [15]). Proposed mechanisms that promote high speciation rates in tropical regions include larger area (a species-area effect[16]), faster evolutionary rate[17] (through the effect of temperature on mutation rate and generation times), and increased biotic interactions[14,18].

Fourth, lower extinction rates in the tropics ("museum of diversity"[14,16]). Stemming from Wallace's work, tropical regions are perceived as more stable and less prone to drastic climate change (e.g., ref. [19]), thereby reducing extinction risk. Further, it has been argued that larger species ranges in tropical areas permit larger population sizes, which also reduce species extinction risk[14].

Tests of these hypotheses have focused primarily on vertebrates and plants. With a few exceptions (e.g., refs. [7,12]) large, densely sampled phylogenetic trees with robust divergence time estimates have been lacking for insects, the most species-rich terrestrial animal group. Here, we generate the first species-level phylogeny of the brush-footed butterflies (Nymphalidae), the most diverse butterfly family (~6400 described species). Over the past two decades, a sustained effort has been made to generate comparable molecular data across the family, such that we can now assemble a densely sampled phylogenetic tree (e.g., refs. [20–23]). We aggregate data from virtually all nymphalid species ever sequenced to date and generate a time-calibrated tree of 2866 species, representing about 45% of the extant described species. Previous studies have already shown that

nymphalid butterflies originated in the Late Cretaceous[20,24] and diversified across all continents. The family exemplifies a latitudinal diversity gradient with ~83% of described species distributed in the Neotropics, Afrotropics, or Southeast Asian biogeographic regions, while the Palearctic and Nearctic regions together account for ~15% of the total species richness (this study).

Here, we assess the relative contribution of time-for-speciation, asymmetry of dispersal events, and variation in net diversification rate (speciation minus extinction) in generating the modern LDG of nymphalids using the time-calibrated phylogeny and biogeographic information. These four mechanisms, however, usually assume a binary model in which processes occurring at the same latitude are homogenous but differ from processes occurring at different latitudes. Yet, the dynamics of diversification and the underlying processes may differ widely among regions for historical reasons (e.g., different colonization times) or geologic and climatic features (e.g., Andean uplift in South America). Accordingly, we investigate the extent to which age, dispersal, speciation, and extinction differ longitudinally across the tropical regions.

Based on the current level of information and methods available, we show that the history of Nymphalidae shares fundamental similarities with other groups, including a strong LDG, a Laurasian origin, a conserved ancestral tropical niche, high Neotropical diversification during the Eocene, and higher diversification in the Palearctic during the Oligocene. However, we also unveiled notable differences with previous LDG studies, in particular, dynamics of dispersal and diversification greatly varied through time and across tropical regions, in contrast to the idea that (based on the evidence from Nymphalidae at least) the LDG resulted from homogeneous diversification processes across all tropical areas.

## Results

**Time-calibrated supertree.** Nymphalidae diverged from its sister clade (Riodinidae + Lycaenidae) *ca.* 93.2 [84.4–101.8] million years (Myr) ago in the Late Cretaceous, and began to diversify *ca.* 84.6 [76.0–91.8] Myr ago (Fig. 1, Supplementary Data 1, Supplementary Methods 1–2). This age is within the range of previously inferred ages for the family. Our source of secondary calibrations[24] found a crown age of 82.0 [68.1–98.3] Myr ago, an estimate similar to refs. [25–27]). Note that[20] found a mean crown age about 12 Myr older. The backbone topology of our tree agreed with previous studies, but the position of Libytheinae was poorly supported. The taxon is often recovered as a sister to all other Nymphalidae. A study with substantially more genetic loci (352 markers) did not increase support for its position within the family[28].

**Biogeographic patterns of global diversification.** Ancestral area estimations with a maximum-likelihood Dispersal-Extinction-Cladogenesis (DEC[29]) model inferred an ancestral range at the root of Nymphalidae covering Southeast Asia, Palearctic and western Nearctic in the Cretaceous (Supplementary Methods 3, Supplementary Figure 3, Supplementary Table 3). We remain cautious about this result because long branches associated with widespread groups such as the Libytheinae and Danainae can be problematic for ancestral range estimations[30]. Nevertheless, early lineages diversified almost entirely in Southeast Asia before they dispersed towards the Afrotropics and Neotropics by the end of the Paleocene (66–56 Myr ago, Fig. 2, Supplementary Figure 45). During the Eocene (56–34 Myr ago), Southeast Asia became even more central to the dispersal of Nymphalidae with nearly 60% of intercontinental dispersal events originating from that region (Fig. 2, Supplementary Figures 4–5; Supplementary Movie 1).

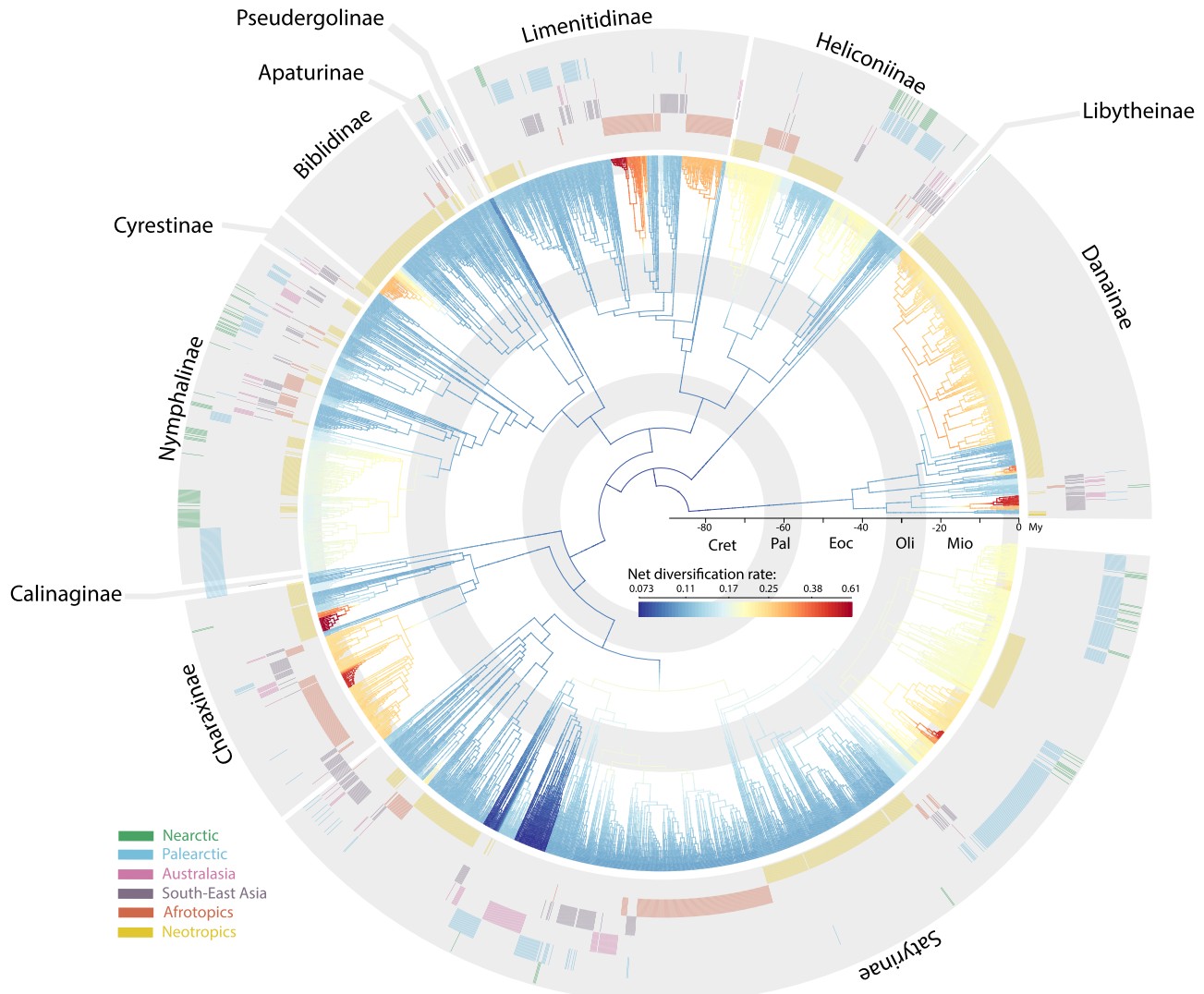

**Fig. 1 Time-calibrated phylogenetic tree of brush-footed butterflies (Nymphalidae) with biogeographic distribution of extant species and branch net diversification rate estimated by BAMM.** Colored bars in outer circles indicate the biogeographic distribution of each terminal taxon included in the tree. Branches are colored according to the average posterior net diversification rate from a birth-death analysis as performed with BAMM. Subfamilies are indicated outside the tree. Gray circles inside the phylogeny indicate geological time periods: Cret= Cretaceous, Pal= Paleocene, Eoc= Eocene, Olig= Oligocene, Mio= Miocene.

Many dispersal events dated to the Eocene occurred between low latitude tropical regions. During the Oligocene (34–23 Myr ago), Southeast Asia remained the most common origin of dispersal events between regions with almost 30% of dispersal events originating from that region (Fig. 2, Supplementary Movie 1). However, during this epoch, we inferred more frequent dispersal events from the Afrotropics into Southeast Asia (*ca*. 10%). Compared to earlier periods, the Neotropics became increasingly isolated, whereas interchanges continued between Southeast Asia, Australasia, the Afrotropics, and the Palearctic, a pattern that was strengthened during the Miocene (Fig. 2). Three types of dispersal events prevailed in the Miocene: from Southeast Asia toward Australasia (*ca*. 24% of the dispersal events—twice as frequent as during the Oligocene), from Southeast Asia toward the Palearctic (*ca*. 18%), and from the Palearctic toward the Nearctic (*ca*. 17%).

The average net diversification rate (speciation minus extinction) across nymphalid butterflies increased through time globally and in all regions except Australasia (Figs. 1, 3, Supplementary Methods 4–6, Supplementary Figures 4–5, Supplementary Movie 1). We did not find any major difference between the

low- and high-latitude regions. Overall, the Afrotropics and Nearctic followed the same trend of monotonic increase of net diversification rate through time. We found an increase in net diversification rate in the Neotropics during the Eocene, which was clearly higher than other tropical regions, but followed a trend similar to the Afrotropics over the last 30 Myr. By contrast, net diversification in Southeast Asia was much slower than the other tropical regions. We found an interesting temporal pattern of diversification in the Palearctic. The Palearctic was characterized by a rapid increase in net diversification rate at the end of the Eocene, the fastest diversification rate of any region during the Oligocene. Speciation and net diversification diminished during a period corresponding to the mid-Miocene climatic optimum (*ca*.15 Myr ago) before increasing again. Australasia showed a distinctly different pattern (Fig. 3). Australasia was the only region characterized by decreasing speciation and net diversification rates. Both were particularly high during the Eocene, but decreased rapidly at the end of the Eocene and continued until the Pliocene. When trying to separate the speciation rate from the extinction rate we found contrasting patterns among the different

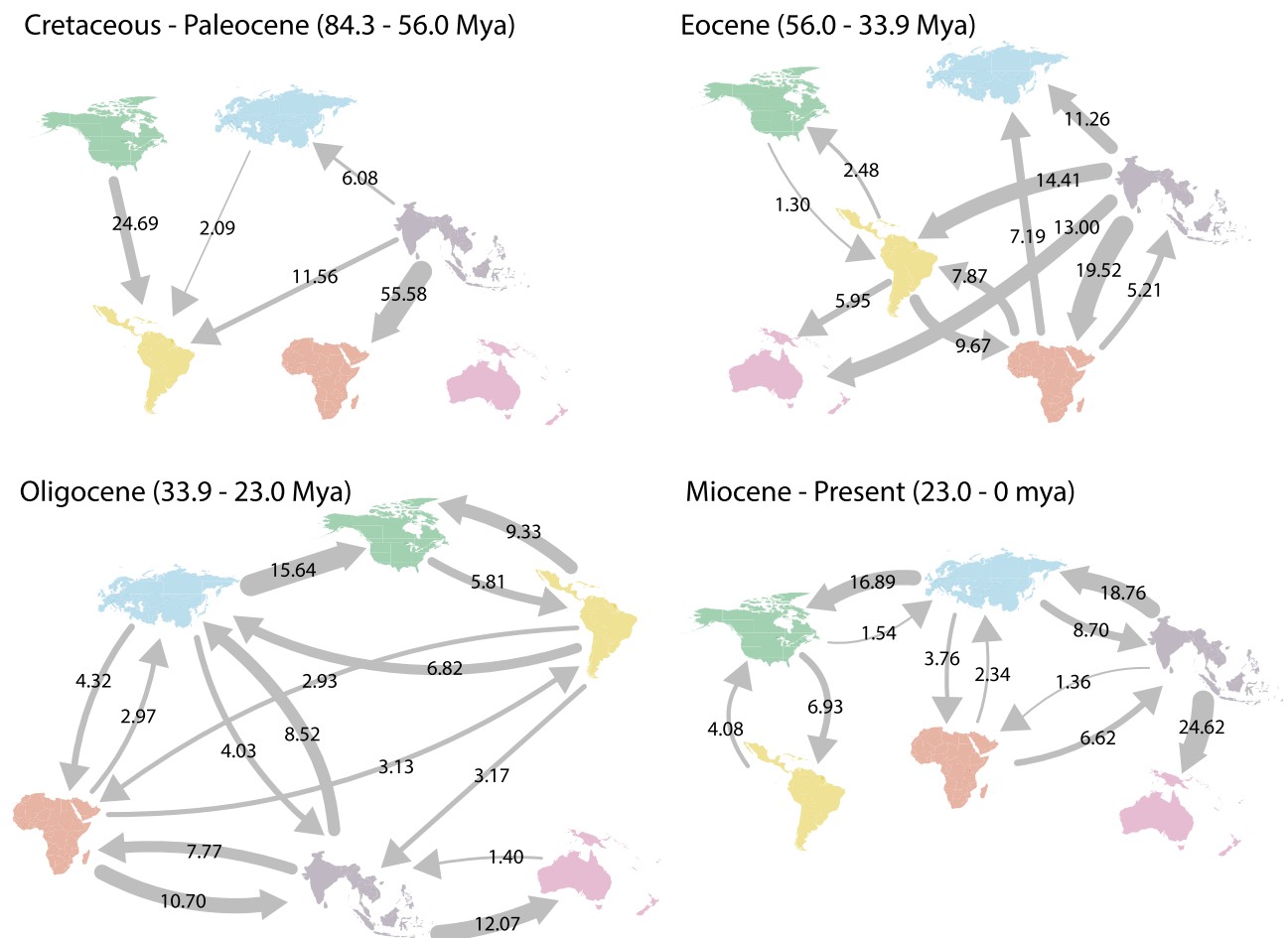

**Fig. 2 Percentages of dispersal events between regions as inferred by our DECX analysis through different geological periods.** Arrows indicate the direction of dispersal and numbers denote the percentage of the total number of events during each period of time. All events representing less than 1% are not shown.

tropical regions. In particular, we found the lowest average extinction rate in the Neotropics during the Eocene, Oligocene, and Miocene.

Comparing the relative frequency of lineages sampled in the tree in each biogeographic region, we found the Eocene to be a period of major transition. Neotropical, Afrotropical, and Australasian lineages increased in relative proportion to Nearctic and Palearctic lineages (Fig. 3). Strikingly, Nearctic and Palearctic lineages reached their lowest historical relative frequency at the end of the Eocene before starting to diversify to their modern extent.

**Biogeographic patterns of regional diversification**. We compared the diversification dynamics among continents, and focused on clades including at least four sampled lineages having mostly diversified in a single region (hereafter called regional diversification events, Supplementary Methods 7). We identified 90 regional diversification events: 30 of these clades are Southeast Asian or Australasian, 21 are Afrotropical, 21 are Neotropical, 14 are Palearctic, and four are Nearctic. We extracted the crown age and inferred net diversification at the crown (netDiv$_{crown}$), net diversification at present (netDiv$_0$), and the time-variation ($\alpha$) of diversification rate from time-dependent birth-death models fitted to each clade. We found almost no significant differences in parameters between regions. The Neotropics had the most species-rich clades but on average, we only found a significant difference with Southeast Asia. Nearctic diversification events were on average the youngest (Fig. 4). Southeast Asian + Australasian nymphalid clades showed

the widest range of crown ages, ranging from 1.37 to 43.87 Ma (Fig. 4). The Neotropics were also characterized by a wide range of crown ages and had, on average, the oldest radiations. The Afrotropics were characterized by the widest range of net diversification parameters at present (netDiv$_0$), while Southeast Asia + Australasia was characterized by the widest range of net diversification rates parameter at the origin of the clade (netDiv$_{crown}$). However, we found no significant difference between regions for any diversification parameter (Fig. 4).

For each biogeographic region we tested which of crown age, netDiv$_{crown}$, netDiv$_0$, and $\alpha$ best predicted the extant species richness of the diversifying clades. We used hierarchical partitioning to compare all possible combinations of parameters and identify the best-fitting model (Supplementary Table 7, Supplementary Figure 7). Models including only the crown age of the radiation consistently explained the largest fraction of the variance in each of the three tropical regions, reaching 85.8% for the Afrotropics. In the Palearctic, crown age explained 53.4% of the variance, but the best-fitting models also included either netDiv0 only, or netDiv0 and $\alpha$ (Supplementary Table 7). In the Nearctic, crown age explained only 27.4% of the variance and the best-fitting models included at least three parameters (Supplementary Figure 7).

## Discussion
**Latitudinal comparisons**. Species richness in Nymphalidae peaks in tropical latitudes, and at least 83% of nymphalid species are

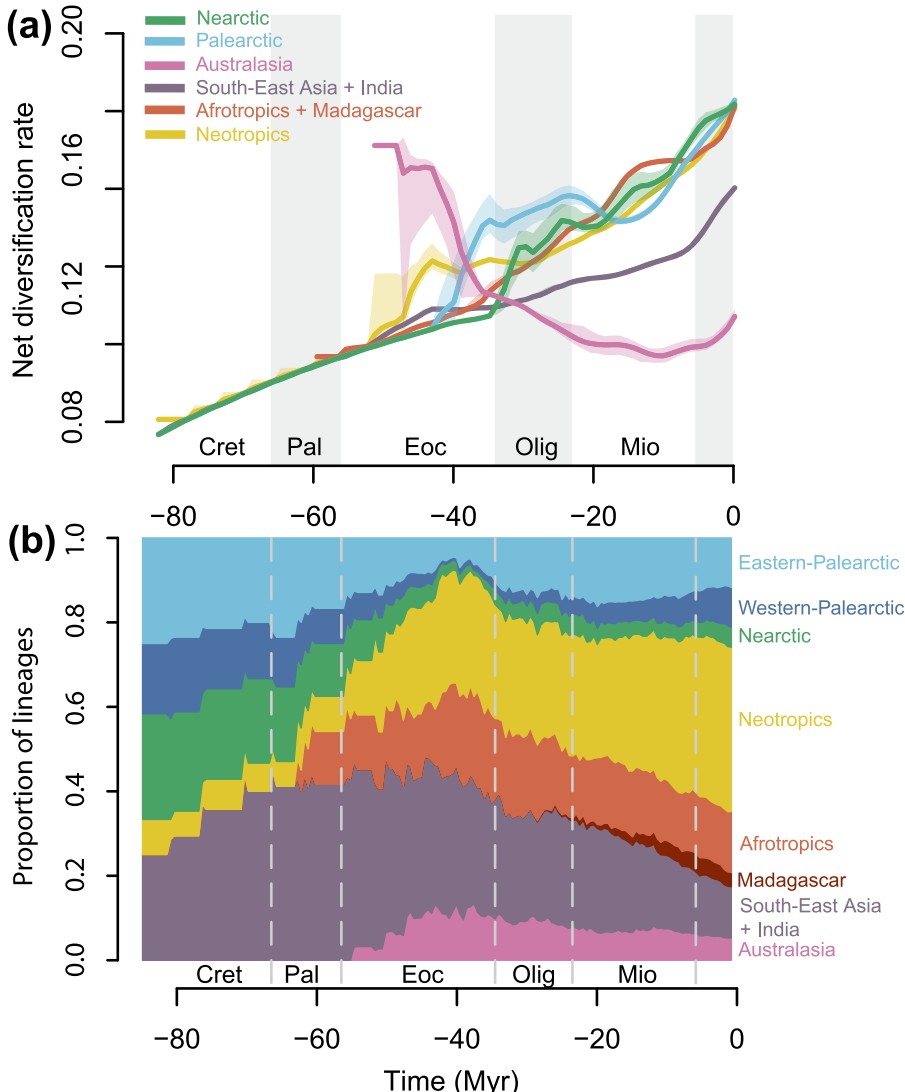

**Fig. 3 Diversification rate and relative proportion of lineages through time in each biogeographic region. a** Average temporal dynamics of net diversification rate in each region and **b** relative proportion of lineages in each biogeographic region through time, estimated from the DECX analysis. Rates were estimated using a sliding window analysis combining historical biogeography (DEC) and speciation/extinction rates (BAMM). Colored shading in figure (**a**) indicates the distribution of mean rates estimated for 100 randomly sampled timing of dispersal events. Colored lines are the mean of this distribution. Cret= Cretaceous, Pal= Paleocene, Eoc= Eocene, Olig= Oligocene, Mio= Miocene. Source data are provided as a Source Data file.

found in the Neotropics, Afrotropics, Madagascar, and Southeast Asia (Figs. 1, 3). We demonstrate that this elevated tropical diversity does not result from simple latitudinal differences in diversification rates. Nymphalid butterfly lineages in tropical regions did not consistently diversify more rapidly than in temperate regions (Figs. 3, 4). The net diversification rate in Southeast Asia was lower than in any other region except Australasia, and we found a high net diversification rate in the Palearctic, especially during the Oligocene. Previous studies on butterflies (Papilionidae[12]) and ants[7] agree with our results and find no consistent latitudinal gradient in diversification rate, refuting the hypothesis that the LDG simply results from latitudinal differences in diversification. Instead, there is increasing evidence that the modern LDG appeared after the Eocene as a result of global climate changes throughout the Cenozoic combined with niche conservatism in tropical lineages (e.g., refs. [2,31]). We argue here that nymphalid butterflies also conform to this scenario.

The Late Cretaceous climate was warmer and less seasonal[32]. Warm and humid conditions seemed to extend to higher latitudes

as documented by fossil faunas (including insects) and floras recovered in either the modern Palearctic or Nearctic (e.g. ref. [33]). For example, ref. [34] found Eocene insect diversity at 50° North paleolatitude to be as diverse as modern tropical diversity. According to our estimation, Nymphalidae arose *ca.* 84 Myr ago in present-day Eurasia and North America (Laurasia). A Laurasian origin has been reported in many different groups, such as the butterfly family Papilionidae[12], palm trees[35], or carnivores[13]. Lineage-area frequency through time shows that until the Eocene, diversity was more evenly distributed between high and low latitudes. These results suggest that nymphalid butterflies (and perhaps all butterflies) were ancestrally adapted to tropical climates and readily dispersed across tropical regions during their earliest period of diversification.

Earth's climate cooled abruptly during the Eocene-Oligocene transition[36]. The appearance of the Antarctic Circumpolar Current strengthened climatic gradients leading to more pronounced seasonality at high latitudes[37]. Fossil evidence indicates extirpation and contraction of "tropical-like" faunas

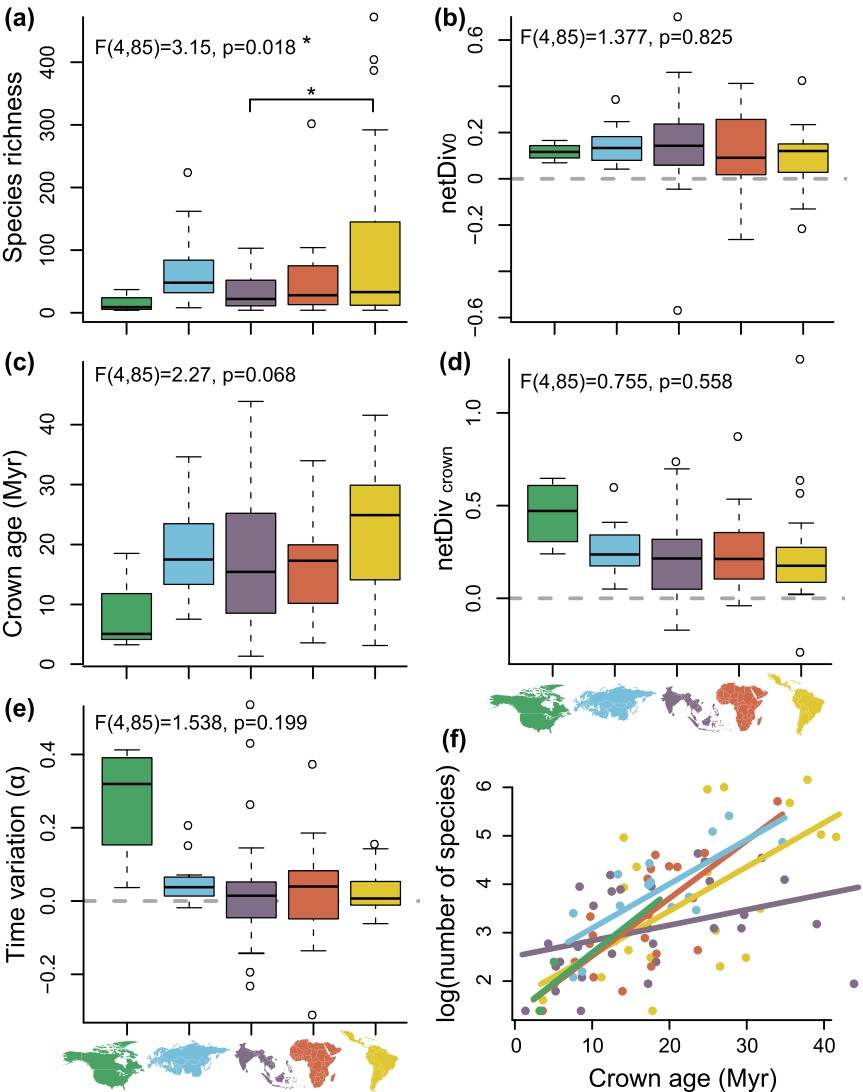

**Fig. 4 Estimated parameter distributions of 90 regional diversification events. a** Extant number of species in diversification events in the different regions. **b** Net diversification rate at the present (netDiv0) in different regions. **c** Crown age of diversification events in different regions. **d** Net diversification at the crown (netDivcrown) in different regions. **e** Time variation parameter (α) in different regions. **f** The log number of extant species within each diversification event regressed against its crown age, analyzed by region. Points, lines, and boxes are colored according to the biogeographic region they represent. Boxplots show the median, interquartile range, default whiskers, and outliers of parameter distributions. The results of a one-way ANOVA performed for every parameter are found above each boxplot. Source data are provided as a Source Data file.

and floras towards equatorial latitudes[31,38] and concomitant ecological turnover[33]. We found that the contributions of the Nearctic and Palearctic fauna to global nymphalid lineage diversity were lowest during the Eocene (Fig. 3). We also found that, despite being colonized shortly after the family evolved, there was a high extinction rate in the Nearctic from the early Oligocene until the mid-Miocene, probably explaining why the Nearctic region only accounts for about 3.5% of extant nymphalid diversity. Hence, early nymphalids probably occupied high latitudes of the Nearctic and eastern Palearctic until the end of the Eocene, but local extirpations and southward contractions accompanying global temperature decline prevented lineages from persisting and diversifying in these regions.

The importance of longer time-for-speciation in tropical areas was also highlighted in the regional diversification analyzes. For the three tropical regions, we found that the crown age of these groups alone explained between 65 and 85% of the species richness variance, and suggested an important effect of clade age.

Additional parameters involving diversification rate did not provide a significantly better fit.

We found an increase in net diversification rate in the Palearctic at the end of the Eocene, and the estimated average diversification rate in the Palearctic was higher than all other regions during the Oligocene. This peak of diversification may have resulted from the emergence of cold-tolerant lineages triggered by the cooler Oligocene climate, as proposed, for example by ref. [39]. This peak of diversification may also have resulted from colonization of the western Palearctic after the Turgai Sea retreated by the end of the Eocene, as suggested for other taxonomic groups (e.g., ref. [40]). Our study also indicates a decrease in net diversification rate during the mid-Miocene climatic optimum before increasing again during the recent Earth cooling[41]. This may reflect temperate-adapted lineages diversifying faster when the climate was cooler and slower during warming events. We found that species richness variation in the Palearctic diversification events are best explained by the

combination of both the age of these radiations and the recent diversification rate. This suggests the importance of recent events that perhaps included glaciations, in shaping the modern Palearctic diversity.

**Longitudinal comparisons**. Our results identify an important role of time-for-speciation, global climate change, and the phylogenetic conservatism of ancestral tropical climatic niches as explanations for modern nymphalid LDG. However, this generalization hides great disparities in the evolutionary histories of tropical regions, each characterized by unique diversification and dispersal histories through time.

Southeast Asia was central for diversification until the end of the Eocene and can be seen as an ancient "cradle of diversity" (Fig. 2, see Supplementary Movie 1 for an animation of the historical biogeography of nymphalid butterflies). The region seems to have been home to most of the Paleocene diversification and was a major source for lineages that dispersed into the Neotropics, Afrotropics, Palearctic, and Australasia. However, net diversification in the region greatly decreased over time relative to the other regions.

In parallel, the Afrotropics show a relatively gradual increase in net diversification rate over time. According to our estimation, the Miocene was the period during which the average net diversification in the Afrotropics was the highest among all regions. The Miocene was characterized by more dispersal events from Africa towards Madagascar, which triggered speciation and multiple endemic radiations (e.g., *Heteropsis*[21]). More importantly, Africa experienced major climate and paleoenvironmental changes throughout this same period[42]. The end of the mid-Miocene climatic optimum initiated a shift from a warm and humid climate associated with trans-African forests to dry (arid) conditions, accompanied by the expansion of savannahs and $C_4$ plants[43,44], probably leading to substantial species turnover.

The Neotropics are currently the most species-rich region and home to at least 37% of extant nymphalid species. Unlike Africa and Southeast Asia, which frequently exchanged lineages, we found that the Neotropics became increasingly isolated over time. We did not detect a particularly high average net diversification rate in the Neotropics, except during the Eocene. This period seems crucial to explaining high Neotropical diversity, and we found that diversification peaked during this period of time compared to the other regions. We also found the Eocene to be the transition period during which the relative proportion of Neotropical diversity approximately doubled between the early and late Eocene (Fig. 3b). Using a deep-time palynological series of Neotropical plants, ref. [45] found that the rate of speciation (and total diversity) peaked during the Eocene, which probably resulted from global warming and the expansion of tropical lineages into higher latitudes. Of note, we found that the average extinction rate was remarkably low throughout the history of Neotropical diversification and that Neotropical clades tended to be on average older than African, Palearctic, and Nearctic clades, although this result was not statistically supported (Fig. 4). Therefore, our results suggest that the combination of early colonizations of the Neotropics, early diversification rate and maybe low extinction rates lead to the steady accumulation of lineages over time, thus supporting the hypothesis that the Neotropics are a "museum of diversity"[46].

Finally, our results indicate that the extant Australasian fauna largely results from multiple dispersal events from Southeast Asia rather than in situ diversification. Indeed, we found a clear pattern of decreasing net diversification rate through time, but a strong increase in dispersal events during the Miocene from Southeast Asia (*ca.* 25% of global dispersal events during this period).

**Study limitations**. Despite the size of our dataset of 2900 species, over 50% of Nymphalidae diversity was not included in our phylogenetic tree. Our dataset corresponds to the molecular information currently available for this group and we included information about missing taxa as much as possible throughout our analyses. Hence, this paper also provides a broad synthesis of the state of Nymphalidae phylogenetics, refining the current taxonomic and geographic sampling gaps and paving the road for future work. Our estimation of divergence times is in line with previous estimates. One of the main unverified sources of potential problems in our phylogenetic reconstruction is saturation, which might affect both taxa relationships and branch length. We did not test for saturation effects that might affect our results especially at the deepest sections of the tree. As far as we know, this issue has never been thoroughly explored for the standard set of genes used in butterfly phylogenies. However, ref. [47] showed that the set of markers used in this study all contain phylogenetic signal and their combination significantly improved the resolution of phylogenetic relationships. In addition, ref. [24] from which secondary time-calibration were taken from, showed that adding or removing the mitochondrial gene fragment (most likely to be affected by saturation problems) did not change the estimation of divergence time at the scale of a backbone of Papilionoidea. According to ref. [48] saturation problems can be partly compensated by increasing the number of taxa and using complex models of substitution. Here, we applied these principles by including as many species as possible and using complex partition strategies identified using Partition-Finder.

It is notoriously difficult to provide a reliable estimation of speciation and extinction rates from phylogenies of extant species (e.g. refs. [49–52]). In particular, estimating extinction rates[53,54] or finding the best model of diversification[55,56] remain areas of intense research. The butterfly fossil record is depauperate[57] and provides little information beyond time-calibrating phylogenies[58]. Therefore, for the time being, birth-death models applied to molecular phylogenies are the best option for improving our understanding of the spatial and temporal patterns of butterfly diversification. We limited our interpretation of speciation and extinction rates separately, and focused on the net diversification process, and combined different approaches (BAMM versus regional diversification) in an attempt to cross validate our inferences with alternative approaches and methods.

Identifying the critical ecological and phenotypic drivers of diversification was beyond the scope of this paper. Other processes not accounted for in this study include the interaction between butterflies and their host plants (e.g., refs. [59,60]). While the level of host specificity varies depending on the taxon considered, the presence of butterfly populations in a locality is limited by the presence of their host plants or by their ability to shift adaptively to new hosts, which may in turn foster diversification. As a result, the evolution of host-plant interactions, notably through the diversification of angiosperms themselves and their dispersal across continents, was most likely key in shaping the modern patterns of Nymphalidae diversity.

## Methods

**Time-calibrated tree**. We inferred a time-calibrated tree of Nymphalidae butterflies using sequence data from the 2866 species for which there was at least one of the following 11 focal gene regions: COI, ArgKin, CAD, DDC, EF1a, GAPDH, IDH, MDH, RpS5, RpS2, wingless (Supplementary Methods 1, Supplementary Data 1). This represents ~45% of the estimated number of species in the family. These sequence data were compiled from published and unpublished studies and subjected to multiple cleaning and verification steps.

We generated the final tree using a tree grafting procedure (Supplementary Methods 1, 2, Supplementary Figure 2). First, we built a backbone tree relying on a dataset of 789 species of Nymphalidae with least six gene fragments available and

11 outgroups. The topology for this backbone was generated with RAxML 8.2.12[61] and time-calibrated using BEAST 1.8.3[62] using a set of 20 secondary calibrations from a recent genus-level, time-calibrated tree of all butterflies[24]. Then, we built species-level trees for 15 subclades, which often corresponded to nymphalid subfamilies. For these trees, we included two outgroups and as many species as possible regardless of the amount of molecular information available. We used PartitionFinder 2.1.1[63] to select partitioning strategies and substitution models for each subclade. For each subset, we set an uncorrelated lognormal relaxed clock (unlinked across partitions, i.e., four independent clocks). We used BEAST 1.8.3[62] to estimate the topology and the relative divergence times for these subclades. Finally, the subclade trees were rescaled using the age of the root estimated by the backbone analysis, and grafted onto the backbone tree. This process was performed on the posterior distributions of both the backbone and the subclades to build a posterior distribution of 1000 grafted trees. We used TreeAnnotator 1.8.3[62] to summarize the tree topology with median node age and 95% credibility interval of each node. Outgroups were removed and the resulting tree was used for all subsequent analyzes.

**Inference of biogeographic history**. We performed a maximum-likelihood estimate of geographic range evolution using the Dispersal-Extinction-Cladogenesis (DEC) model[29] as implemented in an extended C++ version of DEC[64], called DECX[65]. We designed a biogeographic model spanning the evolutionary history of Nymphalidae, starting in the Late Cretaceous (Supplementary Methods 3). We assigned extant species to nine biogeographic regions: western Nearctic, eastern Nearctic, western Palearctic, eastern Palearctic, Neotropics, Afrotropics, India, Southeast Asia, and Australasia. We designed a time-stratified model in which both the adjacency matrices and dispersal matrices varied between time periods (Supplementary Table 3). Time was divided into five time periods: 100–80, 80–60, 60–30, 30–10, 10–0 Myr to account for increasing or decreasing connectivity between biogeographic regions through time.

**Estimation of speciation and extinction rate**. We estimated the temporal dynamics of speciation and extinction rates across our phylogeny using BAMM 2.5[66–68] (Supplementary Methods 4). We accounted for missing species by specifying the sampling fraction at the genus level. The analyzes were run for 50 million generations with four reversible-jump MCMC, sampling parameters every 50,000 generations. The output was then analyzed using the R package BAMMtools[67]. We checked that the MCMC converged with an effective sample size above 600 after we discarded the first 10% of samples as burn-in.

**Biogeographic patterns of diversification: combining BAMM and DEC**. To link biogeography and diversification, the number of possible biogeographic ranges prevented the use of character-state-dependent diversification model, because of the number of parameters that needed to be included. We combined BAMM and biogeographic ancestral state estimation in an attempt to estimate the average diversification rate for each biogeographic area. We combined the speciation and extinction rate estimates for ancestral lineages obtained from BAMM with the biogeographic ranges and timing of dispersal events estimated with DECX to estimate the average net diversification rate for each biogeographic area (Supplementary Methods 5).

Dispersal events: we identified the range with the highest probability at each node. For each dispersal event, we drew 1000 random times of dispersal events along the branches. For each replicate, we recorded the number of dispersal events between biogeographic regions occurring during four geological time periods transformed these sums of events into percentages of the total number of events during that time period. For this analysis we reduced the number of areas from nine to six by combining eastern and western Nearctic into Nearctic, eastern and western Palearctic into Palearctic, India, and Southeast Asia into Southeast Asia; Australasia was kept as a single area (Supplementary Methods 5).

Lineage-area frequency through time: we used DECX results to estimate the frequency of lineages sampled in the tree in each area through time. The number of lineages was computed within 0.5 Myr time intervals. If a dispersal event occurred along a branch, we assumed that it occurred at the branch midpoint.

Biogeography and diversification rates: we estimated variation in speciation and extinction rates through time within each biogeographic region by combining DECX ancestral range estimates and speciation/extinction rate estimates from BAMM (Supplementary Methods 5). We recovered the rates of speciation and extinction through time for all branches using the function dtRates (BAMMtools[67]). We used a sliding window analysis to estimate the mean diversification rates through time for each biogeographic region. We computed the average speciation, extinction, and net diversification rates per region within 4 Myr time windows and shifting the window by 1 Myr. Within each time window, if a lineage occupied an area, the rates estimated for this branch (or fraction of the branch in case of dispersal events) contributed to the average rate of the region. The average was computed by estimating the number of events in one area (rate*branch length) divided by the sum of branch lengths occupying this area during the same time interval. We repeated the analysis for 100 random timings of dispersal events. We report here the net diversification results only (but see Supplementary Methods 5).

Animated historical biogeography: to help visualizing the pattern of historical biogeography we displayed a single realization of historical dispersal events through time on a map with present-day positions of continents for simplicity of implementation. We also displayed at the same time the average net diversification rate through time in regions and the relative frequency of lineages in different regions through time (Supplementary Methods 6, Supplementary Movie 1). The code was adapted from ref. [69].

**Biogeographic patterns of diversification: regional diversification**. While combining BAMM and DECX results provided interesting insights into the general spatial and temporal pattern of diversification it lacked proper statistical support. As an alternative, we investigated diversification by analyzing clades that diversified within a single region (Supplementary Methods 7). We refer to such events as "regional diversification". We arbitrarily defined a regional diversification event as a clade of at least four terminal taxa (i.e., extant taxa included in our tree) that has diversified in a single biogeographic region. Because of the large number of dispersal events happening between some of the areas, we circumscribed fewer regions for this approach: Neotropics, Afrotropics, Southeast Asia combined with Australasia, Palearctic, Nearctic. We identified 90 local diversification events, which represent an estimated number of 5373 species (ca. 86% of all nymphalid diversity), and we estimated net diversification, time-variation of the diversification rate (α), and the age of each diversification event. We tested whether (1) any of these parameters differs between regions, and (2) which parameter best explains the extant diversity of these radiations.

For each regional diversification event, we fitted a model in which speciation rate was modeled as an exponential function of time, while extinction rate remained constant[70] using the R-package RPANDA 1.5[71]. For each clade, we estimated the extinction parameter ($\mu$) and the two parameters for speciation: speciation rate at present ($\lambda_0$) and coefficient of time variation ($\alpha$). Using these parameters, we computed the speciation rate at the crown age ($\lambda_{crown}$), the net diversification at present ($netDiv_0$), and the net diversification rate at the crown age ($netDiv_{crown}$). We tested for a significant difference in species richness, $netDiv_0$, $netDiv_{crown}$, $\alpha$, or crown age between regions using a one-way ANOVA for each parameter, followed by a Tukey test when significant. Then, for each region we used hierarchical partitioning to identify which combination of the four parameters best explains the differences in species richness among radiations of a same region.

**Reporting Summary**. Further information on research design is available in the Nature Research Reporting Summary linked to this article.

## Data availability

All sequences used in our manuscript are available on Genbank. All Genbank accession codes can be found in Supplementary Data 1. Biogeographic distributions used in our analyzes are also available in Supplementary Data 1. The backbone tree and the complete grafted phylogeny are available at: https://doi.org/10.5281/zenodo.5463912. Code for generating the animation of biogeographic history is available at: https://github.com/evogytis/nymphalidae-animation. Source data are provided with this paper.

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

## Acknowledgements

This study would not have been possible without the generous support of amateur and professional lepidopterists around the world, too many to name here; thank you to all for providing specimens and information about life histories of the butterflies. F.L.C is supported by an "Investissements d'Avenir" grant managed by the Agence Nationale de la Recherche (CEBA, ref. ANR-10-LABX-25-01). D.J.L. was supported by grants DEB-1541557 from NSF and WW-227R-17 from the National Geographic Society. R.I.V.W. was supported by Leverhulme Trust emeritus programme. M.E. was supported by an ATIP grant, a grant from the Human Frontier Science Program (RGP0014/2016) and a grant from the French National Research Agency (ANR CLEARWING ANR-16-CE02-0012). E.O.A. was supported by Sigma-Xi (G20100315153261), Center for Systematic Entomology and the Council of the Linnean Society and the Systematics Association for the Systematics Research Fund. S.N. was supported by the Swedish Research Council (2015-04218 and 2019-03441). P.M.M. was supported by a Czech Science Foundation grant (Junior GAČR, 20-18566Y). R.V. was supported by projects PID2019-107078GB-I00 / AEI / 10.13039/501100011033 (Agencia Estatal de Investigación) and 2017-SGR-991 (Generalitat de Catalunya). N.W. acknowledges support by the Swedish Research Council (2015-04441) and a start-up grant from the Department of Biology, Lund University.

## Author contributions

N.C. and N.W. designed the study. The study was enabled by the contribution from N.C., C.P., U.K., P.M.M., K.A.P., M.E., A.D.W., D.J.L., C.M.P., P.D., Z.F.F., S.N., C.M., A.Y.K., K.L.S.B., G.L., I.K., A.Z., E.O.A., R.V., R.I.V.W., S.P.M., C.D.J., C.W.W., A.V.L.F., and N.W. to the data. N.C., N.W., and F.L.C. performed the analyzes. G.D. and N.C. designed and created the animated biogeographic history. N.C., N.W., and F.L.C. wrote the first draft and all authors contributed to the final version of the manuscript.

## Funding

## Competing interests

The authors declare no competing interests

## Additional information

[1]Department of Ecology, Swedish University of Agricultural Sciences, Ulls väg 16, 75651 Uppsala, Sweden. [2]Systematic Biology Group, Department of Biology, Lund University, Lund, Sweden. [3]Gothenburg Global Biodiversity Centre, Gothenburg, Sweden. [4]CNRS, UMR 5554 Institut des Sciences de l'Evolution de Montpellier (Université de Montpellier|CNRS|IRD|EPHE), Place Eugene Bataillon, 34095 Montpellier, France. [5]Vaccine and Infectious Disease Division, Fred Hutchinson Cancer Research Center, Seattle, WA, USA. [6]Museo de Historia Natural, Universidad Nacional Mayor de San Marcos, Lima, Peru. [7]IISER-TVM Centre for Research and Education in Ecology and Evolution (ICREEE), School of Biology, Indian Institute of Science Education and Research Thiruvananthapuram, Thiruvananthapuram, India. [8]Biology Centre of the Czech Academy of Sciences, Institute of Entomology, České Budějovice, Czech Republic. [9]Department of Life and Earth Sciences, Perimeter College, Georgia State University, 33 Gilmer Street, Atlanta, GA 30303, USA. [10]ISYEB, CNRS, MNHN, Sorbonne Université, EPHE, Université des Antilles, 57 rue Cuvier, Paris 75005, France. [11]McGuire Center for Lepidoptera and Biodiversity, Florida Museum of Natural History, University of Florida, Gainesville, FL 32611, USA. [12]City College of New York and Graduate Center, CUNY, New York, NY, USA. [13]National Museum of Natural History, Manila, Philippines. [14]Department of Biological Sciences, University of New Orleans, New Orleans, LA, USA. [15]Department of Zoology, Stockholm University, 10691 Stockholm, Sweden. [16]Australian Museum, 6 College Street, Sydney, NSW 2010, Australia. [17]Universidade Estadual de Campinas, Centro de Biologia Molecular e Engenharia Genética, Av. Candido Rondom, 400, 13083-875 Campinas, SP, Brazil. [18]Nature Education Centre, Jagiellonian University, ul. Gronostajowa 5, 30-387 Kraków, Poland. [19]Departamento de Química y Biología, Universidad del Norte, Barranquilla, Colombia. [20]Institut de Biologia Evolutiva (CSIC-UPF), Barcelona, Spain. [21]Department of Life Sciences, Natural History Museum, London SW7 5BD, UK. [22]Durrell Institute of Conservation and Ecology (DICE), University of Kent, Canterbury CT2 7NR, UK. [23]5 Cummington Street, Department of Biology, Boston University, Boston, MA 02215, USA. [24]Department of Zoology, University of Cambridge, Downing St., Cambridge CB2 3EJ, UK. [25]Smithsonian Tropical Research Institute, Gamboa, Panama. [26]Departamento de Biologia Animal, Instituto de Biologia, Universidade Estadual de Campinas (UNICAMP), 13083-862 Campinas, SP, Brazil. ✉email: chazotn@gmail.com

