## [Peer Review File · Nature Communications]

Reviewers' Comments:

Reviewer #1:

Remarks to the Author:

This contribution is the latest in a series of recent papers analyzing the LDG using very large, global phylogenies. The availability of such data has been a relatively recent development, especially in plants and invertebrates, and thus allows new tests of ideas that have been debated for a long time. This contribution, focusing on a diverse group of butterflies, will be a very welcome and important addition to the literature, because of the scope of the dataset, because of the rigorous and thorough analyses used, and because it focuses on an understudied (relative to mammals and birds) invertebrate group. The results find no simple systematic relationships between speciation, extinction, dispersal underlying the LDG. Rather, these factors vary in complex ways through time, and across different biogeographic regions. The authors also find correlations between species richness and clade age, and that tropical clades are older, indicative that longer history of diversification in the tropics may explain the LDG. I can find little fault with this paper, and I commend the authors on a fine piece of work. I do have a few minor comments to consider.

Comments:

-I feel that the discussion, while interesting, does quite a lot of interpretation of the nuances of different patterns, some may say overinterpretation. There are some main rigorous results here, but then there are a number of post hoc explanations for different patterns. Nearly every rise and fall of one of these rates can be attributed to something happening in the world at that time. I feel it helpful to note these potential drivers, but I'm sure the authors would agree this falls short of a rigorous test, in which an a priori prediction is tested statistically against alternatives. At the same time, discussing these potential drivers can of course be useful, as it can set the stage for the next round of investigation. I would urge the authors to be careful about more clearly separating the results of their formal analyses from the post hoc rationalizing of the patterns they observed. The latter should be presented as potential explanations that can be tested rigorously in the future. In general, streamlining the discussion a bit to focus on the strongest conclusions might be warranted.

-Is it possible to acknowledge/mention other potential drivers of the gradient (namely, ecological regulation seems missing). I don't believe this paper rules out such factors being important.

-I was missing some discussion or caveats about the potential limitations of this analysis, namely limitations to the data available (the phylogeny, which seems very well done overall, but of course is far from complete and immutable) as well as potential limitations to the diversification rate analysis and ancestral state estimation methods. These methods are still relatively new, highly debated, and while they are well worth pursuing at this point, they are likely to improve. I feel these points should be briefly mentioned and discussed, along with any other limitations.

-Line 112: I like the formulation and presentation of these four hypotheses, but are they all "historical" explanations? If there is some causal factor that raises speciation rates in the tropics (e.g. radiation or temperature) that seems like it is an evolutionary explanation rather than historical.

-Line 158: The mechanisms themselves do not make assumptions. Certain (but not all) analytical methods may fit models that are homogeneous within latitude. I guess the point you are trying to make is you are focusing more on longitudinal patterns than has typically been done in other such studies?

-Figures (general)

I have a concern that the figures may not be friendly to color blind readers. While color is very helpful in presenting complex data like this, I worry that figures 1 (outside ring), 2, and 4 use a colormap that may not be visible to colorblind people (although I can't be sure, as I can see color). Particularly, red and green of similar saturation are used, and nothing (line style, marker shape) is used to separate groups in a way redundant with color.

Fig 1:

-“Afrotropic” is misspelled in the legend.

-This is a stylistic issue and may be a personal preference, but it is a bit confusing that the grey areas showing taxonomy, don't always overlap the bars showing geography. I realize the information can be seen with this design, so it is not incorrect, but I found it distracting and I would gently recommend a modified design.

Figure 2:

“Coloured shading indicates the distribution of mean rates estimated for 100 randomly sampled timing of dispersal events. Coloured lines are the mean of this distribution.”

- This is a bit confusing, how does dispersal come in here? This seems to be a plot of diversification in each region. Does the dispersal set the assignments of lineages to geography?

Figure 3: I also found this figure confusing when I first saw it. Because the arrows representing dispersal and the macroevolutionary parameters (λ , μ) are similar in style, it is easy to think all arrows reflect dispersals into and out of each region from/to different sources/destinations. I wonder if there is some way to differentiate these a little bit better. Again, this figure is not erroneous, but I suggest it would be improved if it took a little less thinking to “get it”.

In the legend, “Relative contributions” to what? I'm unclear what λ and μ mean here precisely, is it an average for the region over the whole time course? It would also be nice to show net diversification somehow, since that is a key point of interest in the study.

Figure 4: Should there also be a line showing the overall correlation across all regions?

Reviewer #2:

Remarks to the Author:

To understand the worldwide diversity patterns in the brush-footed butterflies the authors use molecular data and statistical analyses to test several previous hypotheses to explain the latitudinal diversity gradient in the group. They find varying support for different hypotheses around the group, which is a common pattern when these questions are asked across diversity groups of organisms with wide/worldwide distributions.

While I appreciate the effort the authors put into this study, there is not much novel here. The data matrix has a lot of missing data; this is gene-based data for 11 genes, but taxa only had to have data for one gene to be included which is troubling; although there are a large number of taxa they are still missing more than 50% of the diversity of the group; they test several hypotheses and find variable support for each in different parts of the world; there are already major concerns in the systematics community about being able to estimate extinction rates at all from molecular data; the author acknowledge there is a lot of variation/lack of confidence around the dating of many of the major clades they focus on, but this is not strictly accounted for; and if these were not butterflies of another charismatic group I wonder if this paper would even be sent for review.

Although I have no doubt the butterfly community will be interested in this paper, I am not convinced that ecologists/evolutionary biologists in general will be excited about this paper.

Minor Essential Revisions:

The Results section does not contain the usual “results” you expect to find. There are many

statements about this rate when up and this rate went down, but there is no numerical results provided or statistics. This is really just a summary of the results. This should be expanded upon. Most sequences do not have GenBank voucher numbers and these must be provided before publication.

Reviewer #3:

Remarks to the Author:

In this manuscript by Chazot and co-authors, the researchers conducted a very well-done evolutionary analysis of the latitudinal diversity gradient (LDG) in brush-footed butterflies (Nymphalidae), comparing speciation, extinction and dispersal processes over space and time in order to evaluate the generality of four historical hypotheses that could explain greater species richness in tropical regions. The main results, based on a new, large and densely sampled phylogeny of this diverse butterfly family, identify great variation in the dynamics of dispersal and diversification over time and across tropical regions, and that the LDG has not resulted from homogeneous diversification processes across all tropical regions, as has been previously suggested. The authors suggest that global climate change throughout the Cenozoic, particularly during the Eocene-Oligocene transition, combined with the conserved ancestral tropical niches, played a major role in generating the modern LDG of these butterflies.

The results are novel and will they be of interest to others in the areas of biogeography and evolutionary ecology. The novelty is also in the creative way they have analyzed the data, using a mixture of historical biogeographical approaches with estimations of macroevolutionary rates, which makes this work also interesting for a wider field. I enjoyed reading this paper and in general, I do not have major revisions. However, to complete the robustness of the manuscript some clarifications are necessary, as follows:

1.- The authors should show the analyses used to test the strength of the molecular database to perform phylogenetic analyses. The phylogenetic signal of the database can be improved with the quality of primary alignments through a selection of the genes that are least subject to saturation (Philippe et al., 2011). This saturation effect is especially sensitive for molecular clock tree estimations, given that it leads to an underestimation of observed divergence times, particularly for older phylogenetic events (see Wilke et al., 2009). In fact, significant levels of saturation are not suitable for molecular clock estimations. The authors can use the critical saturation value estimated from the index of substitution saturation proposed by Xia et al. (2003), given that its application enables researchers to quickly judge whether a set of aligned sequences is useful in phylogenetics.

2.- Please clarify which molecular clock-based model was used to build the tree, and the criteria to select this model (e.g. strict or relaxed molecular clock models?).

3.- The authors tested for a relationship between crown age, netDiv0, netDivcrown, or α and the estimated total number of extant species (log-transformed species richness) by fitting a linear model for each biogeographic region. However, a simple linear model is not appropriate given the multiple independent variables used, so a multiple regression model is more appropriate. Also, this approach, together with some strategy to find the "best" regression model (e.g. Hierarchical partitioning method), can help to find the smallest subset of predictors that provides the "best fit" to the observed total number of extant species. This "best" subset of predictors should include those that are most important in explaining the variation in the observed total number of extant species.

Philippe H, Brinkmann H, Lavrov DV, Littlewood DTJ, Manuel M, et al. (2011) Resolving Difficult Phylogenetic Questions: Why More Sequences Are Not Enough. *PLoS Biol* 9(3): e1000602. doi:10.1371/ journal.pbio.1000602

Wilke T, Schulthei R, & C Albrecht (2009) As Time Goes by: A Simple Fool's Guide to Molecular Clock Approaches in Invertebrates. *American Malacological Bulletin* 27: 25-45. <https://doi.org/10.4003/006.027.0203>

Xia X, Xie Z, Salemi M, Chen L & Y Wang (2003) An index of substitution saturation and its

application. *Molecular phylogenetics and evolution* 26 (1). 1-7. [https://doi.org/10.1016/S1055-7903\(02\)00326-3](https://doi.org/10.1016/S1055-7903(02)00326-3)

REVIEWER COMMENTS

Reviewer #1 (Remarks to the Author):

This contribution is the latest in a series of recent papers analyzing the LDG using very large, global phylogenies. The availability of such data has been a relatively recent development, especially in plants and invertebrates, and thus allows new tests of ideas that have been debated for a long time. This contribution, focusing on a diverse group of butterflies, will be a very welcome and important addition to the literature, because of the scope of the dataset, because of the rigorous and thorough analyses used, and because it focuses on an understudied (relative to mammals and birds) invertebrate group. The results find no simple systematic relationships between speciation, extinction, dispersal underlying the LDG. Rather, these factors vary in complex ways through time, and across different biogeographic regions. The authors also find correlations between species richness and clade age, and that tropical clades are older, indicative that longer history of diversification in the tropics may explain the LDG. I can find little fault with this paper, and I commend the authors on a fine piece of work. I do have a few minor comments to consider.

>>> Thank you for reviewing our study and the general positive input. Below we have taken into account your comments, which overall improved our study.

Comments:

-I feel that the discussion, while interesting, does quite a lot of interpretation of the nuances of different patterns, some may say overinterpretation. There are some main rigorous results here, but then there are a number of post hoc explanations for different patterns. Nearly every rise and fall of one of these rates can be attributed to something happening in the world at that time. I feel it helpful to note these potential drivers, but I'm sure the authors would agree this falls short of a rigorous test, in which an a priori prediction is tested statistically against alternatives. At the same time, discussing these potential drivers can of course be useful, as it can set the stage for the next round of investigation. I would urge the authors to be careful about more clearly separating the results of their formal analyses from the post hoc rationalizing of the patterns they observed. The latter should be presented as potential explanations that can be tested rigorously in the future. In general, streamlining the discussion a bit to focus on the strongest conclusions might be warranted.

>>> We understand the reviewer's comment and fully agree. Although the models used in historical biogeography are process-based and include paleogeographic scenarios, the pattern emerging from the analyses usually implies post hoc tentative explanations. All scenarios built from our analyses are indeed hypotheses that can be further tested with new data and models. As a consequence, we tone down our interpretations and change the Discussion to better differentiate our results from the conclusions we are proposing.

-Is it possible to acknowledge/mention other potential drivers of the gradient (namely, ecological regulation seems missing). I don't believe this paper rules out such factors being important.

>>> We agree that many factors could be at play to produce such a complex pattern that built up over tens of millions of years (Mittelbach et al. 2007 – *Ecol. Lett.*; Mannion et al. 2014 – *TREE*). Because we primarily focused on the historical biogeography, our results do not rule out any other ecological factor. Accordingly, we focused our Discussion around this historical aspect of the LDG. However, there are definitely other factors underlying diversification (e.g. Wiens et al. 2006 – *Am. Nat.*; Condamine et al. 2012 – *Ecol. Lett.*; Rolland et al. 2014 – *PLoS Biol.*; Pullido-Santacruz & Weir 2016 – *Evolution*; Meseguer & Condamine 2020 – *Evolution*). One of the most obvious drivers in our case would be host-plant interactions, which have been demonstrated as an important factor in the LDG of swallowtail butterflies (Condamine et al. 2012 – *Ecol. Lett.*). Hence, in the revised version, we added a paragraph acknowledging other factors potentially responsible for the nymphalid LDG.

-I was missing some discussion or caveats about the potential limitations of this analysis, namely limitations to the data available (the phylogeny, which seems very well done overall, but of course is far from complete and immutable) as well as potential limitations to the diversification rate analysis and ancestral state estimation methods. These methods are still relatively new, highly debated, and while they are well worth pursuing at this point, they are likely to improve. I feel these points should be briefly mentioned and discussed, along with any other limitations.

>>> We definitely agree that the field of macroevolutionary biology relies on recently developed methods, such as probabilistic birth-death models (Morlon 2014 – *Ecol. Lett.*) or ancestral areas estimation (Ronquist & Sanmartín 2011 – *AREES*). In the last decade, a wealth of approaches has been proposed, tested, and criticized. This study is thus not free from methodological caveats and we have added to the Discussion, a section entirely dedicated to the limitations of the study.

-Line 112: I like the formulation and presentation of these four hypotheses, but are they all “historical” explanations? If there is some causal factor that raises speciation rates in the tropics (e.g. radiation or temperature) that seems like it is an evolutionary explanation rather than historical.

>>> The reviewer is right in that there are causal factors behind higher speciation, as much as there are behind differences in extinction and dispersal among regions. For instance, host-plants need to be already present in a region to be colonized by butterflies. The framework focuses on the signal in speciation and extinction to estimate patterns through time and space, rather than on testing the causal drivers of their variation simply because we currently do not have enough information to tease apart ecological and evolutionary mechanisms at the scale of Nymphalidae. Hence, by “Historical” we imply the study of patterns through time, of dispersal and diversification rate in our case.

-Line 158: The mechanisms themselves do not make assumptions. Certain (but not

all) analytical methods may fit models that are homogeneous within latitude. I guess the point you are trying to make is you are focusing more on longitudinal patterns than has typically been done in other such studies?

>>> We chose to keep regions separated longitudinally because we are interested in quantifying differences across areas that are at the same latitudes. We could have run additional analyses, e.g. SSE models, comparing tropical vs temperate, which is what most studies have been doing (e.g. Rolland et al. 2014 – PLoS Biol.; Pullido-Santacruz & Weir 2016 – Evolution; Meseguer & Condamine 2020 - Evolution), but we feel that this would have increased the amount of analyses and content of the manuscript, which is already dense.

-Figures (general)

I have a concern that the figures may not be friendly to color blind readers. While color is very helpful in presenting complex data like this, I worry that figures 1 (outside ring), 2, and 4 use a colormap that may not be visible to colorblind people (although I can't be sure, as I can see color). Particularly, red and green of similar saturation are used, and nothing (line style, marker shape) is used to separate groups in a way redundant with color.

>>> We checked again our color scheme. According to our tests, the color scheme should be still working for the two most common cases of color-blindness. Only the two less frequent cases (0.59% and 0.56% respectively) would have issues. However, in this case we are also facing the problem of the number of colors. We need seven colors, which greatly limits the possible color schemes.

Fig 1:

-“Afrotropic” is misspelled in the legend.

>>> DONE

-This is a stylistic issue and may be a personal preference, but it is a bit confusing that the grey areas showing taxonomy, don't always overlap the bars showing geography. I realize the information can be seen with this design, so it is not incorrect, but I found it distracting and I would gently recommend a modified design.

>>> We changed the figure to reduce the non-overlapping grey area, i.e. the separation between subfamilies.

Figure 2:

“Coloured shading indicates the distribution of mean rates estimated for 100 randomly sampled timing of dispersal events. Coloured lines are the mean of this distribution.”

- This is a bit confusing, how does dispersal come in here? This seems to be a plot of diversification in each region. Does the dispersal set the assignments of lineages to geography?

>>> Dispersal events estimated along the phylogeny via stochastic simulations are at play because they model the timing when one lineage changes from one biogeographic region to another, hence the assignment of the branch rate within a time window. Each stochastic replicate can also potentially change the source area of

the dispersal event, although in this case it does not affect the region-specific mean diversification rate.

Figure 3: I also found this figure confusing when I first saw it. Because the arrows representing dispersal and the macroevolutionary parameters (λ , μ) are similar in style, it is easy to think all arrows reflect dispersals into and out of each region from/to different sources/destinations. I wonder if there is some way to differentiate these a little bit better. Again, this figure is not erroneous, but I suggest it would be improved if it took a little less thinking to “get it”.

>>> We decided to move this figure to the supplementary information only and provide explanations on how these parameters were estimated.

In the legend, “Relative contributions” to what? I’m unclear what λ and μ mean here precisely, is it an average for the region over the whole time course? It would also be nice to show net diversification somehow, since that is a key point of interest in the study.

>>> The figures and explanations have now been moved to the supplementary information only.

Figure 4: Should there also be a line showing the overall correlation across all regions?

>>> We modified this part of our manuscript following the Reviewer’s 3 advices, with a slightly different focus and statistical tests. We also modified the figure included in the manuscript.

Reviewer #2 (Remarks to the Author):

To understand the worldwide diversity patterns in the brush-footed butterflies the authors use molecular data and statistical analyses to test several previous hypotheses to explain the latitudinal diversity gradient in the group. They find varying support for different hypotheses around the group, which is a common pattern when these questions are asked across diversity groups of organisms with wide/worldwide distributions.

>>> Thank you for reviewing our manuscript, and we feel sorry that you think our study does not bring novel results or ideas. Below we try to convince you the opposite.

While I appreciate the effort the authors put into this study, there is not much novel here. The data matrix has a lot of missing data; this is gene-based data for 11 genes, but taxa only had to have data for one gene to be included which is troubling; although there are a large number of taxa they are still missing more than 50% of the diversity of the group; they test several hypotheses and find variable support for each in different parts of the world; there are already major concerns in the systematics community about being able to estimate extinction rates at all from molecular data; the author acknowledge there is a lot of variation/lack of confidence around the dating of many of the major clades they focus on, but this is not strictly accounted for; and if these were not butterflies of another charismatic group I wonder if this paper would even be sent for review.

>>>With a bit more than 6000 species, Nymphalidae butterflies are comparable to Passeriformes or Mammalia in terms of species richness. We present here a comprehensive time-calibrated phylogenetic tree that is the result of an effort from the butterfly community over the last 20 years. We believe that there is currently no other group of insects that matches our phylogenetic tree in terms of both total number of species and sampling fraction, which makes the current tree one of the most comprehensive datasets outside of the overwhelming vertebrate literature.

Including species with a single gene fragment is not unusual, especially in these large trees (see Jetz et al. 2012 – Nature; Tonini et al. 2016 – Biol. Cons.; Economo et al. 2018 – Nat. Comm.; Upham et al. 2019 – PLoS Biol.). We would like to remind that the tree is not built at once but rather through trees at a smaller phylogenetic scale given the size of our dataset. The topologies of the different subclades and the relative position of species were checked for every subclades and obvious misplacements of taxa (especially the single gene species) identified upstream using Maximum Likelihood tree estimations.

There is no doubt that this phylogenetic tree will be updated in the future as more information becomes available. For this study, we have combined all the information available to date and analyzed it using the state-of-the-art methods. Hence, discussing data limitations of this study highlights the gap existing in 2020, between the amounts of efforts invested in vertebrate phylogenetics and insect phylogenetics (e.g. Troudet et al. 2017 – Sci. Rep.). Ensuring that biodiversity is representatively sampled and studied while this is still possible is an urgent prerequisite for understanding how such a staggering species richness build up through time, and coped with tectonic and environmental changes, and eventually helping for its conservation.

Although I have no doubt the butterfly community will be interested in this paper, I am not convinced that ecologists/evolutionary biologists in general will be excited about this paper.

>>> We respectfully disagree with the reviewer #2's statement here. Studying the macroevolutionary mechanisms of LDG is of interest to a large community, and studying the LDG of an invertebrate group is not common so we think this will be of great value to the community. The LDG of all tetrapod groups has been already studied with phylogenies and/or fossils (sometimes several times) in mammals (e.g. Rolland et al. 2014 – PloS Biol.), birds (e.g. Pullido-Santacruz & Weir 2016 – Evolution), amphibians (Pyron & Wiens 2013 - PRSB), squamates (Pyron 2014 – GEB), turtles (Nicholson et al. 2015 – Nat. Comm.), and crocodiles (Mannion et al. 2015 – Nat. Comm.). In comparison, despite their great species richness, there are only a handful of studies dealing with the invertebrate LDG in a phylogenetic framework, including swallowtail butterflies (Condamine et al. 2012 – Ecol. Lett.), water beetles (Morinière et al. 2016 – Sci. Rep.), and ants (Economo et al. 2018 – Nat. Comm.). More studies on the invertebrate LDG using macroevolutionary approaches are necessary.

Minor Essential Revisions:

The Results section does not contain the usual “results” you expect to find. There are many statements about this rate when up and this rate went down, but there is no numerical results provided or statistics. This is really just a summary of the results. This should be expanded upon.

>>> There is indeed a limited number of statistical tests because of the data and methods that are available to us nowadays. For example, the absence of ecological or phenotypic data compiled at the scale of all Nymphalidae butterflies prevents any tests linking trait evolution and diversification. Our work here focused on the geographical pattern of diversification, through the use of biogeographic ancestral state estimation. Biogeographic ancestral state estimation using DECX (or any other biogeographic framework) does not provide any formal statistical test, for example to test differences in dispersal events between regions. The use of trait-dependent diversification models (e.g. MuSSE or ClaSSE) to link biogeography and diversification is an alternative option that provides a formal test for differences among regions. However, the large number of possible geographic ranges (combinations of areas) prevents the use of such models, since the number of parameters and complexity of the models become too high. Nevertheless, we slightly changed the focus of our analyses of the regional diversification events and included additional statistical tests that are now reported in the manuscript and Supplementary Information. We added in particular an ANOVA test to statistically assess differences in diversification parameters between regions and a hierarchical partitioning to compare the fit of all possible combinations of parameters for each region and estimate the percentage of species richness variance explained by each parameter.

Most sequences do not have GenBank voucher numbers and these must be provided before publication.

>>> Sequences have now been deposited in GenBank and all have a voucher number. The table has been updated.

Reviewer #3 (Remarks to the Author):

In this manuscript by Chazot and co-authors, the researchers conducted a very well-done evolutionary analysis of the latitudinal diversity gradient (LDG) in brush-footed butterflies (Nymphalidae), comparing speciation, extinction and dispersal processes over space and time in order to evaluate the generality of four historical hypotheses that could explain greater species richness in tropical regions. The main results, based on a new, large and densely sampled phylogeny of this diverse butterfly family, identify great variation in the dynamics of dispersal and diversification over time and across tropical regions, and that the LDG has not resulted from homogeneous diversification processes across all tropical regions, as has been previously suggested. The authors suggest that global climate change throughout the Cenozoic, particularly during the Eocene-Oligocene transition, combined with the conserved ancestral tropical niches, played a major role in generating the modern LDG of these butterflies.

The results are novel and will they be of interest to others in the areas of biogeography and evolutionary ecology. The novelty is also in the creative way they have analyzed the data, using a mixture of historical biogeographical approaches with estimations of macroevolutionary rates, which makes this work also interesting for a wider field. I enjoyed reading this paper and in general, I do not have major revisions.

>>>Thank you for the constructive review and your general positive input on our study.

However, to complete the robustness of the manuscript some clarifications are necessary, as follows:

1.- The authors should show the analyses used to test the strength of the molecular database to perform phylogenetic analyses. The phylogenetic signal of the database can be improved with the quality of primary alignments through a selection of the genes that are least subject to saturation (Philippe et al., 2011). This saturation effect is especially sensitive for molecular clock tree estimations, given that it leads to an underestimation of observed divergence times, particularly for older phylogenetic events (see Wilke et al., 2009). In fact, significant levels of saturation are not suitable for molecular clock estimations. The authors can use the critical saturation value estimated from the index of substitution saturation proposed by Xia et al. (2003), given that its application enables researchers to quickly judge whether a set of aligned sequences is useful in phylogenetics.

>>> The gene fragment that is probably the most sensitive to saturation effect in our dataset is the mitochondrial gene fragment (COI). Because of this potential problem we precisely removed this fragment from the dataset used for generating the time-calibrated backbone. Thus, the backbone was only time-calibrated using nuclear gene fragments. The mitochondrial gene fragment was included in the subclades. Of note, Chazot et al. (2019 – Syst Biol) in their time-calibration of a butterfly backbone tree – which was used as the primary source of time-calibration points for our study – compared the results obtained with and without mitochondrial fragments. The results showed a very minor effect on node ages, with slightly younger median age but identical credibility intervals. Therefore, we are confident that the choices we made for dating the tree yielded the least biased results, relative to the saturation problem.

2.- Please clarify which molecular clock-based model was used to build the tree, and the criteria to select this model (e.g. strict or relaxed molecular clock models?).

>>> This information was available in the Supplementary Information only in our previous version of the manuscript. We have now added the basic information in the main document, albeit detailed information remains in the Supplementary Information.

3.- The authors tested for a relationship between crown age, netDiv0, netDivcrown, or α and the estimated total number of extant species (log-transformed species richness) by fitting a linear model for each biogeographic region. However, a simple linear model is not appropriate given the multiple independent variables used, so a multiple regression model is more appropriate. Also, this approach, together with some strategy to find the “best” regression model (e.g. Hierarchical partitioning method), can help to find the smallest subset of predictors that provides the “best fit” to the observed total number of extant species. This “best” subset of predictors should include those that are most important in explaining the variation in the observed total number of extant species.

>>> Thank you for this excellent suggestion. We modified the regional diversification part to follow the reviewer’s suggestions. In the revised version, this part includes two different statistical tests. First, we tested differences between regions in species number, crown age, netDiv0, netDivcrown, and α using ANOVA and post-hoc tests. Second, we used hierarchical partitioning to compare the fit of all possible combinations of parameters for each region and estimate the percentage of species richness variance explained by each parameter. Both the main document and supplementary material have been modified accordingly and figures modified to show the parameter distributions and the results of the statistical tests.

Philippe H, Brinkmann H, Lavrov DV, Littlewood DTJ, Manuel M, et al. (2011) Resolving Difficult Phylogenetic Questions: Why More Sequences Are Not Enough. *PLoS Biol* 9(3): e1000602. doi:10.1371/ journal.pbio.1000602

Wilke T, Schulthei R, & C Albrecht (2009) As Time Goes by: A Simple Fool's Guide to Molecular Clock Approaches in Invertebrates. *American Malacological Bulletin* 27: 25-45. <https://doi.org/10.4003/006.027.0203>

Xia X, Xie Z, Salemi M, Chen L & Y Wang (2003) An index of substitution saturation and its application. *Molecular phylogenetics and evolution* 26 (1). 1-7. [https://doi.org/10.1016/S1055-7903\(02\)00326-3](https://doi.org/10.1016/S1055-7903(02)00326-3)

Reviewers' Comments:

Reviewer #1:

Remarks to the Author:

I have checked the revisions carefully and I feel the authors have addressed all my comments as well as made other improvements. I think the paper is much improved, with a cleaner and consistent message well represented by the figures. I have no further concerns.

Reviewer #2:

Remarks to the Author:

The authors have addressed many of the three reviewers concerns, although they have ignored some of the suggestions for new/additional analyses by myself and Reviewer 3. I still have very serious concerns about the amount of missing data (which they do not address at all) and missing species (which they justify because other studies have missing taxa too) that directly impacts their analytical results and their ability to have confidence in their findings. As I am in the minority in my excitement for this paper I will leave it up to the other reviewers to determine if this meets the criteria for publication in Nature Communications.

Reviewer #3:

Remarks to the Author:

I do not have new comments for the authors. However, regarding the authors state of "confident that the choices we made for dating the tree yielded the least biased results, relative to the saturation problem", I recommend that: Given that this is scientific research it is not enough to have confidence, is necessary show evidence that the database used is really not saturated. It is very important that the authors show the results of saturation analysis (e.g. Xia Test) that could demonstrate not significant saturation in the database. That is because the molecular clock and phylogenetic analyses are really sensitive to the saturation problems, and also to the node effect artefact in branch length estimation. So, I suggest performing the tests that can sustain and give more support to their very interesting results. Please performs the substitution saturation test proposed by Xia et al. (2003), and the node-density effect artefact test by Venditti et al (2006).

Chris Venditti, Andrew Meade, Mark Pagel, Detecting the Node-Density Artifact in Phylogeny Reconstruction, *Systematic Biology*, Volume 55, Issue 4, August 2006, Pages 637–643, <https://doi.org/10.1080/10635150600865567>

Xia X, Xie Z, Salemi M, Chen L & Y Wang (2003) An index of substitution saturation and its application. *Molecular phylogenetics and evolution* 26 (1). 1-7. [https://doi.org/10.1016/S1055-7903\(02\)00326-3](https://doi.org/10.1016/S1055-7903(02)00326-3)

REVIEWER COMMENTS

Reviewer #1 (Remarks to the Author):

I have checked the revisions carefully and I feel the authors have addressed all my comments as well as made other improvements. I think the paper is much improved, with a cleaner and consistent message well represented by the figures. I have no further concerns.

>>> Thank you for your last comments.

Reviewer #2 (Remarks to the Author):

The authors have addressed many of the three reviewers concerns, although they have ignored some of the suggestions for new/additional analyses by myself and Reviewer 3. I still have very serious concerns about the amount of missing data (which they do not address at all) and missing species (which they justify because other studies have missing taxa too) that directly impacts their analytical results and their ability to have confidence in their findings. As I am in the minority in my excitement for this paper I will leave it up to the other reviewers to determine if this meets the criteria for publication in Nature Communications.

>>> Thank you for your new comments. Looking at our taxon sampling, it is important to highlight that all deep lineages are included in our dataset, and missing information is concentrated towards the tip of the tree. When looking at the shallow parts of the phylogeny, at the genus level, of approximately 550 genera described in the family Nymphalidae, 489 are included in our current dataset, i.e. 90%. Of the 59 genera missing, half of them are monospecific genera and altogether these missing genera represent about 2% of nymphalid total species diversity. The general structure of the tree, down to the genus level is therefore very well sampled and most of the missing taxa are therefore distributed below the genus level. Thus, no major gap in the phylogeny of the group exists and the possibility of an important bias associated with that is avoided. We informed our models about missing taxa in every analysis possible, namely BAMM analyses and Morlon et al's models for diversification. In both cases, the sampling fraction was indicated by considering taxonomic placement of missing taxa down to the genus level, hence accounting for them in our analyses.

By mentioning the fact that other studies have missing data/species, we did not intend to build an excuse for our data. Instead, we want to point at the fact that missing taxa in such big phylogenetic trees is currently inherent to this type of studies. We have included about 45% of the species in our tree, hence taxa possessing molecular information, which is a remarkable achievement for an insect group of that size. When looking at, probably, the two most well studied groups: Jetz et al. (2012) “complete” bird phylogeny had molecular information for about 66% of the species, for a total of 3330 missing species. For mammals, Faurby & Svenning (2015) had molecular data for 59% of the species (i.e. 2364 missing taxa), which was improved by Upham et al. (2019) to 69% (i.e. still 1813 missing species). However, we also find other examples with much worse taxon sampling. The phylogeny of Actinopterygian fishes published by Rabosky et al. (2018) in *Nature* included only 36% of species with DNA information, meaning in this case a staggering 19888 missing species. In insects, Economo et al. (2018) published in *Nature Communication* a phylogeny of ants containing 673 species out of 14,512 species, a taxon sampling of only 4.6% ... These studies attempted to compensate for missing taxa by grafting missing taxa on the DNA phylogeny, which only consists in adding taxa as far as taxonomic information helps, otherwise randomly. However, not only these studies were successfully published with large amounts of missing data, but they also further served as phylogenetic framework for many evolutionary studies.

Economo, E. P., Narula, N., Friedman, N. R., Weiser, M. D., & Guénard, B. (2018). Macroecology and macroevolution of the latitudinal diversity gradient in ants. *Nature Communications*, 9(1), 1-8.

Faurby, S., & Svenning, J. C. (2015). A species-level phylogeny of all extant and late Quaternary extinct mammals using a novel heuristic-hierarchical Bayesian approach. *Molecular phylogenetics and evolution*. 84: 14-26.

Jetz, W., Thomas, G. H., Joy, J. B., Hartmann, K., & Mooers, A. O. (2012). The global diversity of birds in space and time. *Nature*, 491(7424), 444-448.

Rabosky, D.L., Chang, J., Title, P.O., Cowman, P.F., Sallan, L., Friedman, M., Kaschner, K., Garilao, C., Near, T.J., Coll, M. and Alfaro, M.E., 2018. An inverse latitudinal gradient in speciation rate for marine fishes. *Nature*, 559(7714), pp.392-395.

Upham, N. S., Esselstyn, J. A., & Jetz, W. (2019). Inferring the mammal tree: species-level sets of phylogenies for questions in ecology, evolution, and conservation. *PLoS biology*, 17(12), e3000494.

Reviewer #3 (Remarks to the Author):

I do not have new comments for the authors. However, regarding the authors state of “confident that the choices we made for dating the tree yielded the least biased results, relative to the saturation problem”, I recommend that: Given that this is scientific research it is not enough to have confidence, is necessary show evidence that the database used is really not saturated. It is very important that the authors show the results of saturation analysis (e.g. Xia Test) that could demonstrate not significant saturation in the database. That is because the molecular clock and phylogenetic analyses are really sensitive to the saturation problems, and

also to the node effect artefact in branch length estimation. So, I suggest performing the tests that can sustain and give more support to their very interesting results. Please performs the substitution saturation test proposed by Xia et al. (2003), and the node-density effect artefact test by Venditti et al (2006).

Chris Venditti, Andrew Meade, Mark Pagel, Detecting the Node-Density Artifact in Phylogeny Reconstruction, *Systematic Biology*, Volume 55, Issue 4, August 2006, Pages 637–643, <https://doi.org/10.1080/10635150600865567>

Xia X, Xie Z, Salemi M, Chen L & Y Wang (2003) An index of substitution saturation and its application. *Molecular phylogenetics and evolution* 26 (1). 1-7. [https://doi.org/10.1016/S1055-7903\(02\)00326-3](https://doi.org/10.1016/S1055-7903(02)00326-3)

>>> We understand the concerns over the reliability of our data and analyses, and see that we should have been clearer in our previous reply to better justify where our confidence in our results comes from. Our confidence relies: (1) on the great amount of studies that have focused on the phylogeny of nymphalid butterflies in the past, enabling the comparison between our work and previous studies and reliable source material, (2) on our dataset containing up to 11 gene fragments, enough to mitigate potential saturation issues with one particular partition. We provide below some detailed explanations.

We used a time-calibrated backbone, with a constrained topology, to which 15 subclades were grafted. For these subclades, topology was not constrained and only relative branch lengths were estimated and then rescaled based on ages estimated in the backbone. Therefore, the reliability of the time calibration lies primarily in the estimates of divergence times of the backbone topology. The backbone encompasses the estimation of the root age, the early divergences and formation of the subfamilies and tribes. Our backbone is certainly not the first estimate of the early diversification of Nymphalidae butterflies. The origin of Nymphalidae butterflies has already been estimated at least seven times before. Chazot et al. (2019) provided a comparison of the root age estimates between these seven studies (Figure 1). Note that all these studies came with different genetic datasets, different sets of time calibration points, priors and phylogenetic scale.

Figure 1. Comparison of node age estimates for Papilionoidea families (here Pieridae, Lycaenidae, Riodinidae and Nymphalidae) from Chazot et al. (2019) study (core analysis) and estimates from previous

studies. Mode and 95% CI for the core analysis are presented. For the other studies the values reported in the original study are used. From Chazot et al. (2019).

All studies showed consistent estimates for the age of Nymphalidae, with variation in confidence intervals resulting primarily from differences in calibration prior information. The broadest variation came from Condamine et al. (2018) who used complete mitochondrial genomes but no nuclear markers and tested different molecular clock strategies. Chazot et al. (2019) provided a comprehensive time calibration of a genus-level phylogeny of Papilionoidea (butterflies) based on a recent revision of the fossil butterflies, complemented with a set of host-plant calibration points. Chazot et al. (2019) also performed extensive testing of divergence time estimate to a variety of model parameters and source data. The results showed that divergence time estimations were robust to many changes in priors and analyses settings (see second figure below). In particular Chazot et al (2019) compared the results obtained from nuclear markers only on one side and the combination of nuclear and mitochondrial markers on the other side (Figure 2). Including mitochondrial gene fragments (particularly prone to saturation problems) did not change the results. Note that in the case of our backbone tree for the present study we included mitochondrial gene fragments but excluded the 3rd codon position, which is probably the partition with the most subject to saturation problem in our entire dataset.

Figure 2. Comparison of node age estimates between the core analysis and the seven alternative analyses for the crown age of the family Pieridae, Lycaenidae, Riodinidae, Nymphalidae. Mode, median, and 95% credibility interval are presented. From Chazot et al. (2019).

We used the estimates provided by Chazot et al. (2019) “core analysis” to time-calibrate the backbone of the Nymphalidae. We think that Chazot et al. (2019) is currently the most up to date and reliable source of secondary time-calibration for butterflies. It is therefore no surprise that the divergence times recovered for the backbone in the present study are consistent with the previously published estimates.

Similarly, the backbone topology did not bring any surprise and the deep relationships among subfamilies and tribes recovered in our tree followed what is already known for more than 10 years. For example, the relative position of the subfamily Libytheinae has always been unclear, including when genome-scale datasets have been used. It has been shown that

branch lengths at the root of the clade Nymphalidae are short and thus alternative topologies at the root have little impact on the estimated times of divergence. Here again, we remain confident that our results are consistent with all evidence accumulated in the past decade.

The effect of saturation highlighted by Xia et al. in 2003 revealed conflicting results between 3rd positions of codons analyzed separately from the 1st or 2nd positions. However, Xia et al. (2003)'s study was published in the context of limited amount of molecular information. As a matter of fact, Xia et al. (2003) exemplified the saturation problem with a single gene, partitioned into the different position. We have up to 11 gene fragments, with the backbone samples precisely chosen to maximize the amount of information. This amount of information largely contributes to alleviate major problems of saturation.

Because a relatively abundant literature lays the foundation of the present study and our comprehensive dataset we are in a position to be confident that our analytical choices and phylogenetic trees are reliable. Additional analyses that would potentially induce small changes in the molecular dataset would imply a massive effort of re-analyses (phylogenetics, diversification and biogeography), which are computationally heavy and cumbersome, and would take more than a year to complete, while we believe they would induce only minor changes in the tree, inconsequential for our conclusions on the history of the group.

Chazot, N., Wahlberg, N., Freitas, A. V. L., Mitter, C., Labandeira, C., Sohn, J. C., Sahoo, R. K., Seraphim, N., de Jong, R., & Heikkilä, M. (2019). Priors and posteriors in Bayesian timing of divergence analyses: the age of butterflies revisited. *Systematic Biology*, 68(5), 797–813. <https://doi.org/10.1093/sysbio/syz002>.

Condamine F.L., Nabholz B., Clamens A.-L., Dupuis J.R., Sperling F.A. 2018. Mitochondrial Phylogenomics, the origin of swallowtail butterflies, and the impact of the number of clocks in Bayesian molecular dating. *Syst. Entomol.* 43:460–480.

Xia X, Xie Z, Salemi M, Chen L & Y Wang (2003) An index of substitution saturation and its application. *Molecular phylogenetics and evolution* 26 (1). 1-7. [https://doi.org/10.1016/S1055-7903\(02\)00326-3](https://doi.org/10.1016/S1055-7903(02)00326-3).

Reviewers' Comments:

Reviewer #1:

Remarks to the Author:

The editor asked for my opinion on the new comments, even though I was satisfied by the last revision. My opinion is that I agree with the authors.

Setting a condition that all species should be on the phylogeny with large amounts of molecular data is necessary to do global analysis like this is unreasonable, and such trees are not even available for vertebrates and that never stopped publication in the highest journals. As the authors mention, some studies have addressed this limitation either by adding more species with limited data per species (like this one), or building very robust backbone trees with limited data, and using modeling and statistical methods to account for missing data toward the tips.

The point of this study is not to resolve the entire butterfly tree of life, it is to test diversification/biogeographic hypotheses on a global scale with available phylogenetic data using the best methods possible to account for various uncertainties, including methods that explicitly deal with missing taxa. Covering 45% of species is actually remarkably high for an insect group like this. Is it possible that, in 30 or 50 years time, when >95% of these butterfly species are sequenced and placed with high certainty, the analysis can be revisited and conclusions could change? Sure. But analyses like these are stepping stones to understanding and science needs first drafts on the way to final drafts. If we wait until the phylogeny and distribution of every species is finished, we might as well shut down biogeography and macroevolutionary biology and come back to it a generation later. The same can be said for the diversification and biogeographic statistical inference methods used here as well, none are perfect and future methods will be better. However, in my opinion it is likely that broad global patterns of diversification are well captured by trees with the current resolution, even if detailed topology toward the tips are uncertain, because the overall structure of the tree is likely to be stable. The current study makes significant advances, and even if there is more to do in the future, it is deserving of publishing in Nature Communications in my opinion.

Regarding the saturation test, it would have most directly addressed the concern of reviewer 3 if they had just done the test. However, it is unlikely saturation is a problem (even if present in COI) since the deeper parts of the backbone where saturation occurs depend on many loci that will not be saturated, and the tree is not starting from scratch because it is building on a body of prior knowledge. So, even if the mitochondrial marker is saturated it should not affect the deeper parts of the tree, and the authors show their dating and topology is consistent with previous studies. More generally, I agree with the authors comments that even if some methodological choice about inclusion of different markers could be improved--and that is subjective at this point--redoing the entire workflow would be months of work and is not justified unless there is a reason to suspect it would fundamentally change the results of the analysis. Tweaking the phylogenetic methods would more likely make small changes to the tree that would have no impact on the downstream analyses and conclusions, but take an enormous effort.

Reviewer #3:

Remarks to the Author:

I agree that the authors have addressed many of the three reviewers concerns, and although they have ignored some of the suggestions for new/additional analyses, at least I hope that the authors discuss in his new version the potential saturation effect. Again, the fact that none of the previous research measured the potential saturation effect, this not mean that this not exist. In fact, under challenging conditions (poor taxonomic sampling, or fast-evolving species), these convergences can be mistaken for true phylogenetic signal, thereby creating systematic errors. However, I understand that reanalyses all database is a titanic work that potentially will not change the main findings of this paper, but I insist to encourage to the authors to discuss in his new version the potential saturation effect. Regarding the potential problem to use the Xia test, they can use a posterior predictive saturation analysis like Lartillot et al. (2007) suggest in the PhyloBayes software. However, this last suggestion is not mandatory, and I will leave it up to the editor to

determine if this meets the criteria for publication in Nature Communications.

Lartillot, N., Brinkmann, H. & Philippe, H. Suppression of long-branch attraction artefacts in the animal phylogeny using a site-heterogeneous model. *BMC Evol Biol* 7, S4 (2007).
<https://doi.org/10.1186/1471-2148-7-S1-S4>

REVIEWERS' COMMENTS

Reviewer #1 (Remarks to the Author):

The editor asked for my opinion on the new comments, even though I was satisfied by the last revision. My opinion is that I agree with the authors.

Setting a condition that all species should be on the phylogeny with large amounts of molecular data is necessary to do global analysis like this is unreasonable, and such trees are not even available for vertebrates and that never stopped publication in the highest journals. As the authors mention, some studies have addressed this limitation either by adding more species with limited data per species (like this one), or building very robust backbone trees with limited data, and using modeling and statistical methods to account for missing data toward the tips.

The point of this study is not to resolve the entire butterfly tree of life, it is to test diversification/biogeographic hypotheses on a global scale with available phylogenetic data using the best methods possible to account for various uncertainties, including methods that explicitly deal with missing taxa. Covering 45% of species is actually remarkably high for an insect group like this. Is it possible that, in 30 or 50 years time, when >95% of these butterfly species are sequenced and placed with high certainty, the analysis can be revisited and conclusions could change? Sure. But analyses like these are stepping stones to understanding and science needs first drafts on the way to final drafts. If we wait until the phylogeny and distribution of every species is finished, we might as well shut down biogeography and macroevolutionary biology and come back to it a generation later. The same can be said for the diversification and biogeographic statistical inference methods used here as well, none are perfect and future methods will be better. However, in my opinion it is likely that broad global patterns of diversification are well captured by trees with the current resolution, even if detailed topology toward the tips are uncertain, because the overall structure of the tree is likely to be stable. The current study makes significant advances, and even if there is more to do in the future, it is deserving of publishing in Nature Communications in my opinion.

>>> Thank you for your very positive input in our manuscript. We are delighted to read you enjoy and support publication of our study.

Regarding the saturation test, it would have most directly addressed the concern of reviewer 3 if they had just done the test. However, it is unlikely saturation is a problem (even if present in COI) since the deeper parts of the backbone where saturation occurs depend on many loci that will not be saturated, and the tree is not starting from scratch because it is building on a body of prior knowledge. So, even if the mitochondrial marker is saturated it should not affect the deeper parts of the tree, and the authors show their dating and topology is consistent with previous studies. More generally, I agree with the authors comments that even if some methodological choice about inclusion of different markers could be improved--and that is subjective at this point--redoing the entire workflow would be months of work and is not justified unless there is a reason to suspect it would fundamentally change the results of the analysis. Tweaking the phylogenetic methods would more likely make small changes

to the tree that would have no impact on the downstream analyses and conclusions, but take an enormous effort.

>>> We are extremely grateful to the reviewer for dedicating time to our manuscript. We fully concur with the reviewer's opinion and greatly appreciate that these opinions are being shared.

Reviewer #3 (Remarks to the Author):

I agree that the authors have addressed many of the three reviewers concerns, and although they have ignored some of the suggestions for new/additional analyses, at least I hope that the authors discuss in his new version the potential saturation effect. Again, the fact that none of the previous research measured the potential saturation effect, this not mean that this not exist. In fact, under challenging conditions (poor taxonomic sampling, or fast-evolving species), these convergences can be mistaken for true phylogenetic signal, thereby creating systematic errors. However, I understand that reanalyses all database is a titanic work that potentially will not change the main findings of this paper, but I insist to encourage to the authors to discuss in his new version the potential saturation effect. Regarding the potential problem to use the Xia test, they can use a posterior predictive saturation analysis like Lartillot et al. (2007) suggest in the PhyloBayes software.

However, this last suggestion is not mandatory, and I will leave it up to the editor to determine if this meets the criteria for publication in Nature Communications.

Lartillot, N., Brinkmann, H. & Philippe, H. Suppression of long-branch attraction artefacts in the animal phylogeny using a site-heterogeneous model. *BMC Evol Biol* 7, S4 (2007). <https://doi.org/10.1186/1471-2148-7-S1-S4>

>>> We are thankful to the referee for spending time reviewing our manuscript. The referee is right in that saturation effects have never been formally tested (at least at the scale of our tree) and therefore we cannot be entirely sure that it does not affect the trees. Accordingly, we have added a short discussion about such an effect in our manuscript as follows:

“Our estimation of divergence times is in line with previous estimates. We did not test for saturation effects that might affect our results especially at the deepest section of the tree. However, we used a concatenation of 11 gene fragments for generating the time-calibrated backbone tree, which helps mitigating a potential saturation effect. In addition, Chazot et al. (2019) from which secondary time-calibration were taken from, showed that adding or removing the mitochondrial gene fragment (most likely to be affected by saturation problems) did not change the estimation of divergence time at the scale of a backbone of Papilionoidea.”